# *circAMN1*-Mediated Ferroptosis Regulates the Expulsion of Placenta in Trophoblast Cells

**DOI:** 10.3390/antiox13040451

**Published:** 2024-04-11

**Authors:** Chen Lv, Lusha Guo, Yue Wang, Zongshuai Li, Xingxu Zhao, Yong Zhang

**Affiliations:** 1College of Veterinary Medicine, Gansu Agricultural University, Lanzhou 730070, China; zoo_monkeys@163.com (C.L.); gls19990709@163.com (L.G.); 18065198306@163.com (Y.W.); 2Gansu Key Laboratory of Animal Generational Physiology and Reproductive Regulation, Lanzhou 730070, China; lizsh2010@163.com; 3State Key Laboratory of Grassland Agro-Ecosystems, Key Laboratory of Grassland Livestock Industry Innovation, Ministry of Agriculture and Rural Affairs, Grassland Agriculture Engineering Center, Ministry of Education, College of Pastoral Agriculture Science and Technology, Lanzhou University, Lanzhou 730020, China

**Keywords:** circAMN1, miR-205_R-1, SLC39A8, ferroptosis, retained placenta, programmed cell death

## Abstract

After delivery, the death of trophoblast cells can promote the expulsion of the placenta. Ferroptosis, an iron-dependent programmed cell death, is involved in mammalian development. Circular RNAs are associated with placental development; however, it is unclear whether circular RNAs regulate the expulsion of fetal membranes through ferroptosis. The gene expression profiles in the tail vein blood of Holstein cows with normal and retained placentas were investigated using RNA sequencing and a GSE214588 dataset. *circAMN1* and *SLC39A8* expression was significantly downregulated in the blood of cows with a retained placenta, whereas *miR-205_R-1* expression was significantly upregulated. We validated erastin-induced ferroptosis in trophoblast cells. Transfection with si-circAMN1 and miR-205_R-1 mimic reduced intracellular total iron, Fe^2+^, and glutathione disulfide levels; increased intracellular glutathione levels and glutathione/glutathione disulfide; and enhanced cell viability in these cells. In contrast, transfection with pcDNA3.1 circAMN1 and an miR-205_R-1 inhibitor promoted ferroptosis. As an miR-205_R-1 sponge, circAMN1 regulated the expression of SLC39A8 to control erastin-induced ferroptosis and regulated the proliferation, invasion, and migration of trophoblast cells. Our findings provide a theoretical basis for studying the mechanism by which programmed cell death regulates fetal membrane expulsion and indicate its potential as a therapeutic target for placenta retention.

## 1. Introduction

The separation between the maternal and fetal placentas after fetal delivery is a prerequisite for the smooth expulsion of the placenta [1,2]. During the expulsion of the placenta, the ability to induce the proliferation, invasion, and migration of trophoblast and endometrial epithelial cells is weakened, increasing cell death and causing the fetal cotyledon to detach from the maternal uterine caruncle, thereby promoting placental expulsion [3]. Research has found that, compared to pregnancy, a large number of apoptotic factors immediately appear in the placenta after fetal expulsion. During normal placental expulsion, placental cell death increases, whereas in cows with a retained placenta (RP), an increase in the placental cell count is observed during the prenatal period [4]. This indicates that the death of placental cells plays a crucial role in the normal separation and expulsion of membranes. It is known that ferroptosis is a programmed cell death (PCD) pattern that facilitates the development of organisms [5,6] and is characterized by the accumulation of a large, oxidative, toxic, labile iron pool, which induces the generation of intracellular lipid reactive oxygen species (ROS); excessive oxidative stress leads to lipid peroxidation and subsequent cell death [7], which can inhibit the invasive ability of trophoblast cells [8]. Erastin is an efficient inducer of ferroptosis, mainly through the inhibition of cysteine/glutamate antiporter activity, ultimately leading to cell death [7]. However, information on the involvement of ferroptosis in regulating the expulsion of fetal membranes is lacking.

Circular RNAs (circRNAs) and micro RNAs (miRNAs) are involved in regulating the process of cell ferroptosis at different developmental stages, as well as regulating cell invasion and migration abilities [9,10,11]. For example, regulating miR-30b-5p/SLC7A11 plays a crucial role in ferroptosis in pre-eclampsia [9]. Circle_0008433 knockdown suppresses the angiogenesis, migration, and proliferation of endometrial stromal cells and promotes apoptosis [10]. Circle_0000848 is an miR-6768-5p sponge that regulates the apoptosis, invasion, and migration of trophoblast cells and, thus, helps control the development and function of the placenta [11]. Therefore, elucidating the mechanism by which circRNA regulates placental expulsion through ferroptosis would be a novel strategy for treating RP.

In this study, we aimed to identify candidate differentially expressed (DE) genes related to PCD in cows with normal fetal membrane discharge (NC) or RP based on previous research [1], RNA sequencing (RNA-Seq), GSE214588 dataset, and bioinformatic analysis. Furthermore, we aimed to elucidate the mechanisms associated with ferroptosis in trophoblast cells in vitro. Our results provide insights into the promotion of placenta expulsion by ferroptosis, which can help develop novel methods for treating RP.

## 2. Materials and Methods

### 2.1. Animals

Holstein cows (age 3–4 years; 500 ± 10 kg; 2–3 parities) without any diseases were chosen as experimental animals (Zhangye, China) and allocated to two groups: normal fetal membrane discharge (NC; *n* = 3) and no expulsion of the fetal membrane within 12 h after delivery (RP; *n* = 3). Caudal vein blood and placental tissue samples were labeled NC-1–3 and RP-1–3, respectively.

The animal experiments were approved by the Animal Protection Committee of Gansu Agricultural University (Lanzhou, China; approval No.: GSAU-Eth-LST-2021-003). All institutional and national guidelines for the care and use of laboratory animals were followed.

### 2.2. RNA Sequencing

Total RNA was extracted and purified from NC and RP blood samples using TRIzol reagent (Thermo Fisher Scientific, Waltham, MA, USA), according to the manufacturer’s instructions. The circRNA libraries were generated using the Ribo-Zero rRNA Removal Kit (Illumina, San Diego, CA, USA), according to the manufacturer’s protocol. Screened circRNAs with *p* < 0.05 and |log2 fold-change| > 1 were considered DE genes [12]. The Targetscan and miRanda databases were used to predict the targeting relationship between genes [13]. The functions of these genes were assessed using Gene Ontology (GO) and Kyoto Encyclopedia of Genes and Genomes (KEGG) using DAVID 6.7 (https://david.ncifcrf.gov/summary.jsp, accessed on 30 October 2021).

### 2.3. Quantitative PCR Analysis (qPCR)

The cDNA was synthesized for qPCR using the Evo M-MLV RT Kit with gDNA Clean (Accurate Biotechnology Co., Ltd., Changsha, China). A miRNA first-strand cDNA synthesis kit (Accurate Biotechnology Co., Ltd.) was subsequently used for miRNA reverse transcription. The qPCR was conducted using the 2× SYBR green qPCR Master Mix (Bimake, Houston, TX, USA) and a LightCycler 96 Real-Time System (Roche, Basel, Switzerland); *GAPDH* and *U6* levels were used for normalization. Each experiment of qPCR was repeated three times, and relative expression was calculated using the 2^−ΔΔCt^ method [14]. The primer sequences of all genes are shown in Appendix A.

### 2.4. Cell Lines

Trophoblast cells were donated by Prof. Longfei Xiao from the Beijing University of Agriculture and screened for *Mycoplasma* contamination using a *Mycoplasma* staining assay kit (C0296; Beyotime, Shanghai, China). Cells were cultivated in Dulbecco’s Modified Eagle Medium/Nutrient Mixture F-12 supplemented with 10% fetal bovine serum (Thermo Fisher Scientific) in a humidified incubator at 37 °C and 5% CO_2_. These cells were treated with erastin (5, 10, 15, 20, 30, and 50 µM) or 10 µM Fer-1 (Selleck, Houston, TX, USA), with 0.2% dimethyl sulfoxide (DMSO) as the negative control.

The circAMN1-overexpressing plasmids (pcDNA3.1 circAMN1), silencing RNA (si-circAMN1), and their negative controls (NC; pcDNA3.1 NC and si-NC) were constructed by GenePharma (Shanghai, China). The miR-205_R-1 mimic/inhibitor and NC were constructed by Sangon Biotech (Shanghai, China). These plasmids and RNA were transfected into trophoblast cells at different concentrations using Lipofectamine 2000 (Thermo Fisher Scientific), according to the manufacturer’s protocol. The RNA sequences are presented in Appendix A.

### 2.5. EdU Assay

EdU was diluted with cell culture medium at a ratio of 1:500 to achieve a final concentration of 20 μM. Cells were removed from the culture medium and incubated with 1 mL of diluted EdU at 37 °C for 2 h. Cell proliferation was analyzed using the BeyoClick EdU-488 Cell Proliferation Detection kit (C0071S; Beyotime), according to the manufacturer’s protocol. The corresponding cell fluorescence was detected using a fluorescence microscope (ECHO, Chicago, IL, USA).

### 2.6. JC-1 and ROS Staining

After cell transfection, the fluorescent probe DCFH-DA (S0033S; Beyotime) was diluted with serum-free culture medium at a ratio of 1:1000, and the fluorescent probe JC-1 (C2003S; Beyotime) was diluted with JC-1 staining buffer at a ratio of 1:200, both resulting in a final concentration of 10 μM. Cells were removed from the culture medium and incubated with 1 mL of diluted DCFH-DA or JC-1 at 37 °C for 20 min to detect intracellular ROS or mitochondrial membrane potential, respectively. The corresponding cell fluorescence was detected using a fluorescence microscope (ECHO).

### 2.7. Measurement of Total Iron, Fe^2+^, GSH, and Glutathione Disulfide (GSSG) Levels

Total iron and Fe^2+^ contents in trophoblast cells were measured using a Cell Total Iron Colorimetric Assay Kit (E-BC-K880-M; Elabscience, Wuhan, China) and Cell Ferrous Iron Colorimetric Assay Kit (E-BC-K881-M; Elabscience), respectively, and a ReadMax 1900 microplate reader (Shanghai Flash, Shanghai, China) was used to measure absorbance at 593 nm. GSH and GSSG concentrations were detected using a GSH and GSSG Assay Kit (S0053; Beyotime) and by measuring the cell absorbance at 412 nm.

### 2.8. Wound-Healing Assay

Transfected cells were cultured in a 6-well plate to 90% confluence. Subsequently, scratch wounds were created using a 200 μL plastic pipette tip. After 24 h of cultivation, the cell culture was washed with fresh culture medium to remove non-adherent cells. Images of the cell culture at 0 h and 24 h were obtained under a fluorescence microscope (ECHO). The migration distance of cells was determined by migration rate = (0 h scratch width − 24 h scratch width)/0 h scratch width.

### 2.9. Immunofluorescence

Cells were treated with 4% paraformaldehyde for 30 min, permeabilized with 0.5% Triton X-100 for 10 min, and blocked with 5% bovine serum albumin for 30 min. The cells were incubated overnight at 4 °C with rabbit anti-ZIP8 (No. 20459-1-AP, 1:400; Proteintech, Rosemont, IL, USA), and then with the secondary antibody fluorescein-conjugated affinipure goat anti-rabbit IgG (H + L) (No. SA00003-2, 1:500; Proteintech) for 1.5 h at 37 °C. To detect the nuclei, all samples were treated with 4′,6-diamidino-2-phenylindole (Beyotime). The results were then visualized using an ECHO fluorescence microscope.

### 2.10. Cell Viability Assay

Cell suspensions were inoculated in a 96-well plate, and 10 μL of Cell Counting Kit-8 (CCK-8) solution (BS350A; Biosharp, Hefei, China) was added before incubation for 2 h. A ReadMax 1900 microplate reader (Shanghai Flash) was used to measure absorbance at 450 nm.

### 2.11. Subcellular Localization

The nuclei and cytoplasm of isolated cells were extracted using NE-PER Nuclear and Cytoplasmic Extraction Reagents (Thermo Fisher Scientific), according to the manufacturer’s protocol. Nuclei and cytoplasmic RNA were extracted using TRIzol reagent (Thermo Fisher Scientific) according to the manufacturer’s instructions. *circAMN1* expression levels were analyzed by qPCR, according to the methods described in Section 2.3.

### 2.12. RNase R Treatment

Total RNA (1 μg) was incubated with 4 U/μg RNase R (R7092S; Beyotime) at 37 °C for 30 min. The reaction was terminated by incubation at 70 °C for 10 min. *AMN1* and *circAMN1* mRNA expression levels were detected using 15% polyacrylamide gel electrophoresis.

### 2.13. Dual-Luciferase Reporter Assay

Dual-luciferase reporter vectors for wild- and mut-type circAMN1 and SLC39A8 were constructed by AZENTA Life Sciences (Nanjing, China). These vectors were co-transfected into trophoblast cells with miR-205_R-1 mimic or mimic NC and incubated for 12 h. Luciferase activity was detected using the Dual-Luciferase Reporter Assay Kit (Promega, Madison, WI, USA), according to the manufacturer’s protocol.

### 2.14. Transmission Electron Microscopy

Trophoblast cells were presoaked in a 2.5% glutaraldehyde solution and then fixed in 1% osmium tetroxide at 4 °C for 2 h. The samples were dehydrated by gradient alcohol and gradient acetone, and double-stained with uranyl acetate-lead citrate. Randomly selected fields of view were photographed under a transmission electron microscope (Hitachi HT7700, Tokyo, Japan).

### 2.15. Transwell Assay

Following transfection, the cells were seeded into Transwell chambers with an 8 μm pore size (Corning, New York, NY, USA) for migration assays, with inserts that were pre-coated with Matrigel (BD Pharmingen, San Jose, CA, USA) for invasion assays. After incubation at 37 °C for 12 h, the cells were treated with 4% paraformaldehyde for 15 min before staining with 0.1% crystal violet. Images of the migrated or invaded areas were recorded in three randomly selected areas using a fluorescence microscope (ECHO).

### 2.16. Quantification and Statistical Analysis

Data are presented as mean and SD. Statistical comparisons were performed using Student’s *t*-test with GraphPad Prism version 9.0 (GraphPad Software, Boston, MA, USA). A *p*-value of *p* < 0.05 was considered statistically significant; *p* < 0.01, highly significant; and *p* > 0.05, not significant.

## 3. Results

### 3.1. Construction of the circAMN1/miR-205_R-1/SLC39A8 Signal Network Axis Related to Ferroptosis

A total of 85 DE circRNAs (31 upregulated, 54 downregulated) were found in the RP vs. the NC. Twelve DE circRNAs were also randomly selected for qPCR validation, and the circRNA expression was consistent with the RNA-Seq data trend (Figure 1A–D and Appendix A).

Using previous research [1], the datasets of GSE214588 were downloaded from the Gene Expression Omnibus (GEO) (https://www.ncbi.nlm.nih.gov/gds/?term=, accessed on 18 October 2022), RNA-Seq data, and the biological characteristics of PCD were used to identify DE genes in these datasets. We focused on the involvement of DE miRNA target genes in apoptosis, cell differentiation, multicellular development, the positive regulation of cell surface and cell population proliferation, the negative regulation of cell apoptosis, chromatin, and the membrane (Figure 1E). These eight biological processes were associated with 18 DE bta-miRNAs (6 upregulated, 12 downregulated). The miR-205_R-1 participated in all eight biological processes and was upregulated 2.1-fold in the RP group compared to the NC group (Figure 1F,G and Appendix A).

The miR-205_R-1 target circRNAs and mRNAs were predicted using the TargetScan and miRanda databases. A total of 11 mRNAs were detected (3 upregulated, 8 downregulated) (Figure 1H). Additionally, 20 circRNAs were identified, of which 7 were downregulated and 13 were upregulated (Figure 1I). By analyzing the DE target genes of miR-205_R-1 using KEGG, we found that the most enriched PCD pathways included MAPK, cancer, RAS, PI3K-Akt, VEGF signaling, and ferroptosis (Figure 1J). The DE target genes of miR-205_R-1 in these eight signaling pathways included EPAS1, VEGFA, and SLC39A8. Only SLC39A8 was negatively correlated with miR-205_R-1, enriched in the ferroptosis pathway, and significantly downregulated by 2.61-fold in the RP group compared with the NC group (Appendix A). Among the DE circRNAs targeted by miR-205_R-1, there were 5 ciRNAs and 15 circRNAs (4 downregulated, 11 upregulated). The circRNAs that were negatively correlated with miR-205_R-1 were circRNA1228, circRNA2480, circRNA3998, and circRNA4015. Among them, the circRNA2480 was significantly downregulated by 3.88-fold in the RP group compared with the NC group (Appendix A). The circRNA2480 corresponded to the host gene, *AMN1*, on chromosome 5 and was formed by the reverse splicing of five exons to form a closed-loop structure with a total length of 546 nt, known as *circAMN1*. Therefore, *circAMN1* was selected as a candidate gene to construct a competitive endogenous RNA (ceRNA) network for subsequent experiments.

The ceRNA signaling network was axis-mediated, PCD–ferroptosis pathway in Holstein dairy cows was constructed by screening the DE target genes, *circAMN1* and *SLC39A8*, with *miR-205_R-1* as the central axis. The qPCR showed that *SLC39A8* and *circAMN1* mRNA expression was significantly lower, whereas *miR-205_R-1* mRNA expression was significantly higher in the blood and placental tissues of the RP group compared to that of the NC group (Figure 1K–M).

### 3.2. Erastin Induces Ferroptosis in Trophoblast Cells

We investigated the effect of ferroptosis on trophoblast cells in vitro. To establish the optimal concentration for inducing ferroptosis, the cells were exposed to erastin at various concentrations. Erastin increased trophoblast cell death and reduced cell proliferation in a dose-dependent manner (Figure 2A–C). Significant cell death was observed at a concentration of 30 µM; therefore, this concentration was used in the subsequent experiments. After erastin treatment, intracellular total iron and Fe^2+^ accumulated, mitochondrial membrane potential weakened, the GSH level and GSH/GSSG level decreased, and GSSG and ROS content increased (Figure 2D–L). Treatment with 10 µM Fer-1 reversed these effects.

### 3.3. miR-205_R-1 Regulates Erastin-Induced Ferroptosis

When the miR-205_R-1 mimic was transfected into trophoblast cells at different concentrations, *miR-205_R-1* mRNA expression increased in all groups (Figure 3A). Based on these findings, 50 nM miR-205_R-1 was used in the subsequent experiments. The miR-205_R-1 mimic promoted cell migration (Figure 3B). Immunofluorescence (IF) showed that the miR-205_R-1 mimic inhibited the fluorescence brightness of SLC39A8 (Figure 3C). Furthermore, the miR-205_R-1 mimic increased cell viability, GSH level, and the GSH/GSSG level and decreased total iron, Fe^2+^, and GSSG levels in erastin-treated cells (Figure 3D–I).

Trophoblast cells were treated with different concentrations of the miR-205_R-1 inhibitor. Based on the decreased *miR-205_R-1* mRNA expression in all groups (Figure 3J), the 50 nM miR-205_R-1 inhibitor was used in the subsequent experiments. In erastin-treated cells, treatment with the miR-205_R-1 inhibitor decreased cell migration (Figure 3K); increased SLC39A8 expression (Figure 3L); reduced cell viability; decreased GSH level and the GSH/GSSG level; and increased total iron, Fe^2+^, and GSSG levels (Figure 3M–R).

### 3.4. circAMN1 Regulates Erastin-Induced Ferroptosis

To investigate the effect of circAMN1, we transfected trophoblast cells with different concentrations of pcDNA3.1 circAMN1. Overall, *circAMN1* mRNA expression significantly increased in all transfected groups (Figure 4A); however, 4 ng was selected as the optimal treatment concentration for the subsequent experiments. pcDNA3.1 circAMN1 reduced cell migration (Figure 4B), and IF analysis revealed that pcDNA3.1 circAMN1 promoted SLC39A8 expression (Figure 4C). Furthermore, pcDNA3.1 circAMN1 decreased the viability of erastin-treated cells, decreased the GSH level and GSH/GSSG level, and increased total iron and Fe^2+^ levels. The level of GSSG increased, but not significantly (Figure 4D–I).

We transfected trophoblast cells with varying concentrations of si-circAMN1 #1 or #2. No knockdown effect was conferred by 10 nM si-circAMN1, whereas 30 and 50 nM si-circAMN1 significantly inhibited *circAMN1* mRNA expression (Figure 4J). Ultimately, 30 nM si-circAMN1 #2 was selected for the subsequent experiments. si-circAMN1 promoted cell migration (Figure 4K), and the IF analysis showed that si-circAMN1 inhibited SLC39A8 expression (Figure 4L). In erastin-treated cells, si-circAMN1 promoted cell viability; increased the GSH level and GSH/GSSG level; and decreased total iron, Fe^2+^, and GSSG levels (Figure 4M–R).

### 3.5. Targeting the Relationship among circAMN1, SLC39A8, and miR-205_R-1

Following transfection with the miR-205_R-1 inhibitor, *circAMN1* and *SLC39A8* mRNA expression significantly increased (Figure 5A), whereas transfection with the miR-205_R-1 mimic had the opposite effect (Figure 5B). Following transfection with si-circAMN1, *miR-205_R-1* mRNA expression significantly increased, whereas that of *SLC39A8* mRNA decreased significantly (Figure 5C). Following transfection with pcDNA3.1 circAMN1, *miR-205_R-1* mRNA expression decreased significantly, whereas that of *SLC39A8* mRNA increased significantly (Figure 5D).

In trophoblast cells, *circAMN1* was primarily expressed in the cytoplasm (Figure 5E). We evaluated the stability of *circAMN1* and linear *AMN1* using RNase R. The circular structure (*circAMN1*) showed higher resistance to RNase R and could amplify cDNA, whereas the linear structure (*AMN1*) could not (Figure 5F).

We identified potential binding sites between circAMN1 and miR-205_R-1 to elucidate the mechanism by which circAMN1 transcriptionally regulates miR-205_R-1 expression. To confirm this interaction, we mutated the circAMN1-binding site, ATGAAGG. The corresponding dual-luciferase reporter assays showed that the mutated circAMN1 could not bind to miR-205_R-1 (Figure 5G).

We identified potential binding sites between SLC39A8 and miR-205_R-1 to elucidate the corresponding regulatory mechanism. To confirm this interaction, we mutated the SLC39A8-binding sites, GGTG and AATGAAGG. The corresponding dual-luciferase reporter assays showed that the mutated SLC39A8 could not bind to miR-205_R-1 (Figure 5H).

### 3.6. circAMN1 Acts as an miR-205_R-1 Sponge to Regulate Erastin-Induced Ferroptosis

We conducted rescue experiments in trophoblast cells to investigate the regulatory effect of circAMN1/miR-205_R-1/SLC39A8 on cell ferroptosis. Next, we examined the proliferation, ferroptosis-related factors, migration, and invasion changes of trophoblast cells treated with erastin, circAMN1, and miR-205_R-1.

EdU and JC-1 staining showed that erastin reduced cell proliferation and mitochondrial membrane potential, whereas the miR-205_R-1 inhibitor exacerbated the process. The si-circAMN1-rescued erastin induced cell proliferation and mitochondrial membrane potential, but the miR-205_R-1 inhibitor reversed the effect of si-circAMN1 on trophoblast cells (Figure 6A,B). Erastin increased the *SLC39A8* mRNA expression level, and during this process, si-circAMN1 decreased the *SLC39A8* mRNA expression level (*p* < 0.01). However, the inhibitor reversed this change (Figure 6C). The miR-205_R-1 inhibitor reduced the changes in GSH and GSH/GSSG and increased the content of GSSG, total iron, and Fe^2+^ (*p* < 0.01, Figure 6D–H). When erastin was used to transfect si-circAMN1 into trophoblast cells, ROS levels decreased, but the miR-205_R-1 inhibitor reversed the effect of si-circAMN1 on cells (Figure 6I). After erastin treatment of cells, mitochondrial cristae decreased or disappeared. During this process, si-circAMN1 completed the structure of mitochondria, but the miR-205_R-1 inhibitor reversed this change (Figure 6J). Erastin reduced the migration and invasion ability of cells, and during this process, si-circAMN1 rescued the migration and invasion ability of cells. However, the miR-205_R-1 inhibitor reversed this change (Figure 6K).

EdU and JC-1 staining showed that erastin reduced cell proliferation and mitochondrial membrane potential, and the mimic rescued the process. However, pcDNA3.1 circAMN1 reversed the effect of the miR-205_R-1 mimic on cell proliferation and mitochondrial membrane potential (Figure 7A,B). Erastin increased the *SLC39A8* mRNA expression level, and during this process, the miR-205_R-1 mimic reduced the *SLC39A8* mRNA expression level (*p* < 0.01). However, pcDNA3.1 circAMN1 reversed this change (Figure 7C). pcDNA3.1 circAMN1 reduced the changes in GSH and GSH/GSSG, and increased the content of GSSG, total iron, and Fe^2+^ (*p* < 0.01, Figure 7D–H). When erastin was used to transfect the miR-205_R-1 mimic into trophoblast cells, ROS levels decreased, but pcDNA3.1 circAMN1 reversed the effect of the miR-205_R-1 mimic on cells (Figure 7I). After erastin treatment of cells, mitochondrial cristae decreased or disappeared. During this process, the miR-205_R-1 mimic restored the integrity of the mitochondrial structure, but pcDNA3.1 circAMN1 reversed this change (Figure 7J). Erastin reduced the migration and invasion ability of cells, and during this process, the miR-205_R-1 mimic rescued the migration and invasion ability of cells. However, pcDNA3.1 circAMN1 reversed this change (Figure 7K).

## 4. Discussion

The mammalian uterus and placenta undergo cell remodeling during each cycle; hence, systematic cell renewal is a fundamental event of uterine and placental cell change, including PCD and cell renewal caused by cell proliferation [15]. Maturation and shedding of the placenta are inseparable from PCD [3]. To elucidate the molecular mechanism by which PCD regulates the shedding of the placenta in Holstein cows, we constructed a circAMN1/miR-205_R-1/SLC39A8 signaling network axis related to PCD using the RNA-Seq and GSE214588 databases. The miR-205 is associated with apoptosis and proliferation, and miR-205_R-1 was named based on the missing base from the right of miR-205 [16,17,18]. Chen et al. [16] found that miR-205-5p/PTK7 regulates the proliferation, invasion, and migration ability of colorectal cancer cells. Inhibition of miR-205 expression promotes Rho family GTPase 3 activity in cardiac ischemia/reperfusion injury, regulating the function of the heart and mitochondria and, thus, reducing oxidative stress and apoptosis [18]. These results indicate that miR-205_R-1 is closely related to PCD; therefore, it was selected as a candidate gene that promotes the expulsion of the placenta.

Expulsion of the placenta results from the PCD of endometrial epithelial and trophoblast cells after placental maturation, reducing cell migration and invasion, weakening the ability of the fetal cotyledonary to grasp the uterus, and promoting the detachment of the fetal lobe from the uterine caruncle. Ferroptosis, a type of PCD, plays a vital role in biological development, especially in the expulsion of fetal membranes [3]. In this study, KEGG pathway analysis revealed that the miR-205_R-1 target gene, *SLC39A8*, was ultimately involved in the ferroptosis pathway. SLC39A8, also known as ZRT/IRT-like protein 8, is a transmembrane, divalent, metal transporter protein originally recognized as a zinc importer [19]. However, it was later found to transport iron with high affinity, thereby influencing the iron ion level and metabolism in vivo [20,21]. In this study, SLC39A8 expression was significantly reduced in the RP group, which may have weakened the ability of the body to transport iron ions. We speculate that, after fetal delivery, the mature placenta and uterus have less iron ion accumulation and trophoblast cells and endometrial epithelial cells cannot undergo ferroptosis, thereby preventing the expulsion of the placenta and causing RP.

Erastin can induce intracellular GSH depletion and ROS accumulation, leading to ferroptosis by inhibiting system xc^−^ activity [22]. Therefore, in this study, we used erastin to construct a ferroptosis model and detected ferroptosis-related indicators. We observed a dose-dependent effect of erastin on trophoblast cell death, with increases in intracellular levels of total iron, Fe^2+^, and ROS. Excessive accumulation of ROS can induce cell damage, which is closely related to GSH [23]. GSH, as a substrate for glutathione peroxidase and glutathione transferase, can eliminate harmful free radicals (mainly oxygen radicals) and lipid peroxides in the body by converting them into fatty acids and water as it is oxidized to GSSG [24]. In this study, erastin induced increases in intracellular GSSG levels and decreases in GSH levels and GSH/GSSG. The depletion of GSH and increase in GSSG levels indicate a decrease in the intracellular ability to clear lipid peroxides, leading to ferroptosis. Oxidative stress causes cell ferroptosis, which can lead to mitochondrial damage, resulting in the reduction or disappearance of mitochondrial cristae and a decrease in mitochondrial membrane potential [25]. In the current study, a decrease in mitochondrial membrane potential induced by erastin was detected. Fer-1 is a lipophilic iron death inhibitor that clears ROS through a reduction reaction, inhibits the Fenton reaction by reducing unstable iron in cells, and regulates ferroptosis [26]. The current study showed that Fer-1 reversed the changes induced by erastin. These results confirmed that a trophoblast ferroptosis model was constructed using erastin.

The non-coding RNAs (ncRNAs) play vital roles in various forms of PCD [27,28,29]. For example, lncRNA Mir9-3hg suppresses cardiomyocyte ferroptosis in ischemia–reperfusion mice via the Pum2/PRDX6 axis [27]. Furthermore, MiR-134-3p targets HMOX1 to inhibit ferroptosis in granulosa cells of sheep follicles [29]. In this study, regulating the expression of circAMN1 or miR-205_R-1 changed the migration ability of trophoblast cells. The si-circAMN1 or the miR-205_R-1 mimic inhibited erastin-induced ferroptosis. In contrast, pcDNA3.1 circAMN1 or the miR-205_R-1 inhibitor promoted erastin-induced ferroptosis. This indicates that circRNAs and miRNAs have regulatory effects on cells and are closely involved in the ferroptosis of trophoblast cells.

To demonstrate the regulatory relationship between circAMN1, miR-205_R-1, and SLC39A8, the current study confirmed the binding sites between circAMN1 and miR-205_R-1 and between miR-205_R-1 and SLC39A8 using dual-luciferase reporter assays. The circAMN1 was primarily expressed in the cytoplasm of trophoblast cells, rather than in the nucleus. It was also found that circAMN1 had a circular characteristic and was not easily degraded by the exonuclease, RNase R. These results indicate that circAMN1 regulates the expression of SLC39A8 through a ceRNA mechanism. In the subsequent results, in erastin-treated cells, si-circAMN1 was transfected into the trophoblast cells, reducing cell ferroptosis and eliminating the effect of the miR-205_R-1 inhibitor. pcDNA3.1 circAMN1 was transfected into trophoblast cells, exacerbating cell ferroptosis and thus eliminating the effect of the miR-205_R-1 mimic. In short, circAMN1 acted as an miR-205_R-1 sponge, regulating the expression of SLC39A8 and erastin-induced in vitro ferroptosis. SLC39A8 mainly transports iron ions, which are essential metal elements involved in several biological processes, from oxygen transportation to protein synthesis and enzymatic reactions [30]. Abnormalities in iron metabolism, including increased iron absorption and decreased iron output, can lead to ferroptosis by generating iron-mediated ROS via the Fenton reaction [31]. Extracellular trivalent Fe^3+^ forms a complex with transferrin, which binds to transferrin receptor 1 on the cell membrane and enters the cell through endocytosis to form an endosome. Fe^3+^ is subsequently reduced to Fe^2+^ by six-transmembrane epithelial antigen of prostate 3 and transported to the cytoplasm through SLC39A8, where it is introduced into a labile iron pool [19]. SLC39A8 can also transport extracellular free Fe^2+^ to intracellular, labile iron pools [19]. Through the Fenton reaction, free iron interacts with H_2_O_2_ to form highly reactive hydroxyl radicals [8]. The H_2_O_2_ and superoxide radicals are decomposed through chain reactions, producing a large amount of molecular oxygen, hydroxyl radicals, and hydroxyl anions, promoting lipid peroxidation in the cell membrane and resulting in ferroptosis progression [31], thereby leading to the reduced invasion and migration ability of endometrial epithelial and trophoblast cells and causing fetal membrane shedding.

To the best of our knowledge, this is the first study to emphasize the regulatory effect of circAMN1 against erastin-induced ferroptosis and to demonstrate its potential as an miR-205_R-1 sponge to regulate SLC39A8 expression, thereby mediating iron transport and regulating ferroptosis induced by erastin in trophoblast cells. Furthermore, the observed synergistic effects of circAMN1/miR-205_R-1 in the context of erastin-induced ferroptosis provide insights into potential biomarkers and present a novel treatment strategy for RP.

## Figures and Tables

**Figure 1 antioxidants-13-00451-f001:**
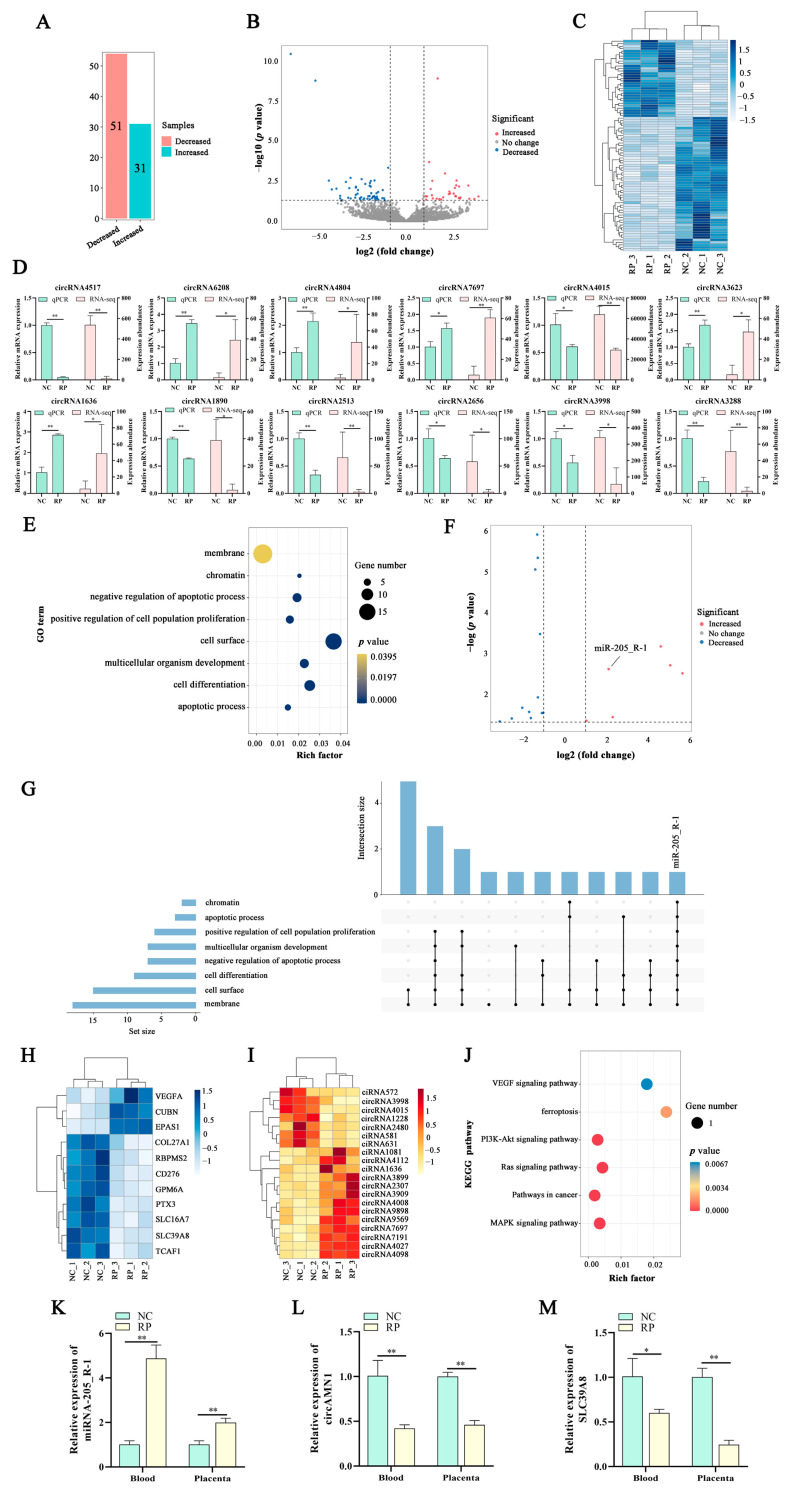
Construction of the ceRNA network axis. (**A**) Number of DE circRNAs. (**B**) Volcano plot of DE circRNAs. (**C**) Heatmap of DE circRNAs; colors from blue (high) to white (low) represent circRNA expression levels (Z-score). (**D**) Comparison of qPCR and RNA-seq of DE circRNAs. (**E**) Gene Ontology functional annotation of DE miRNA target gene mRNAs. (**F**) Volcano plot of DE miRNAs associated with PCD. (**G**) UpSet plot of DE miRNA target genes associated with PCD. (**H**) Heatmap of *miR-205_R-1* target gene mRNAs associated with PCD; colors from blue (high) to white (low) represent mRNA expression levels (Z-score). (**I**) Heatmap of *miR-205_R-1* target gene circRNAs; colors from orange (high) to white (low) represent circRNAs expression levels (Z-score). (**J**) KEGG pathway enrichment of *miR-205_R-1* target gene mRNAs. (**K**–**M**) *circAMN1*, *SLC39A8*, and *miR-205_R-1* mRNA expression in blood and placental tissue of the NC and RP groups. Data are expressed as mean (SD); * represents *p* < 0.05; ** represents *p* < 0.01; competitive endogenous RNA (ceRNA); programmed cell death (PCD); differentially expressed (DE); normal fetal membrane discharge (NC); retained placenta (RP).

**Figure 2 antioxidants-13-00451-f002:**
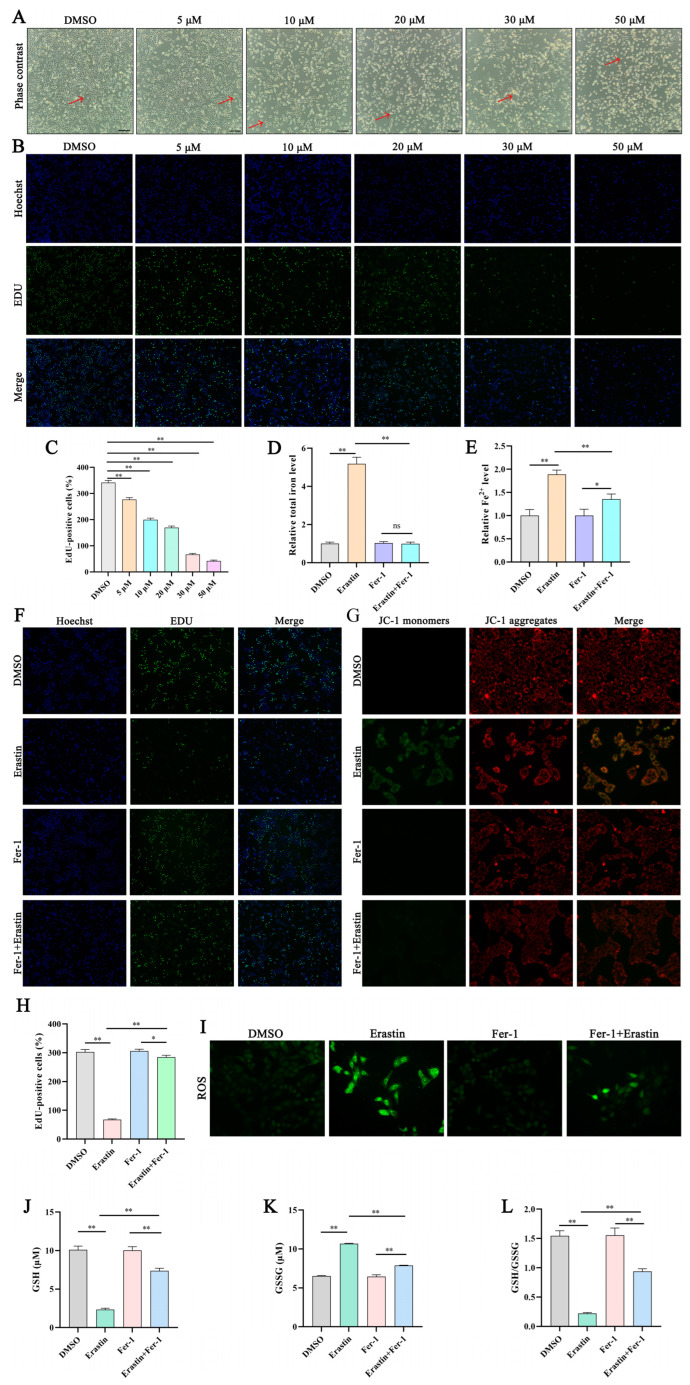
Erastin induces ferroptosis in trophoblast cells. (**A**,**B**) Phase contrast images and EdU assay of trophoblast cells treated with the indicated concentrations of erastin (5, 10, 15, 20, 30, and 50 µM) or DMSO in six-well plates for 12 h or evaluated; 40× magnification; red arrows indicate dead cells. (**C**) Cell proliferation detected using the EdU assay. (**D**–**L**) Trophoblast cells treated with 30 μM erastin or 10 μM Fer-1 for 12 h. (**D**,**E**) Total iron and Fe^2+^ levels in trophoblast cells. (**F**–**H**) Cell proliferation detected using the EdU assay and mitochondrial membrane potential; EdU 40× magnification; JC-1 200× magnification. (**I**) Reactive oxygen species 200× magnification. (**J**–**L**) GSH and GSSG levels and GSH/GSSH. Data are presented as mean (SD); * represents *p* < 0.05; ** represents *p* < 0.01; ns: *p* > 0.05.

**Figure 3 antioxidants-13-00451-f003:**
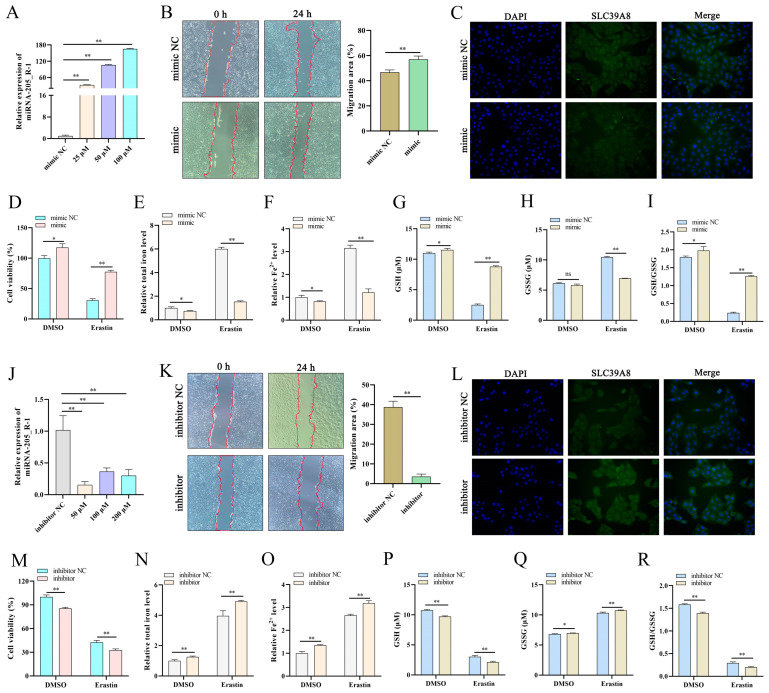
miR-205_R-1 regulates erastin-induced ferroptosis in trophoblast cells. (**A**) Transfection efficiency of an miR-205_R-1 mimic at different concentrations (25, 50, and 100 μM) in trophoblast cells. (**B**,**K**) Wound-healing assay of trophoblast cells transfected with 50 nM miR-205_R-1 mimic (**B**) or inhibitor (**K**), imaged 0 and 24 h following wound infliction. The migration area was calculated using ImageJ; 40× magnification. (**C**,**L**) Immunofluorescence assay of trophoblast cells transfected with the miR-205_R-1 mimic (**C**) or inhibitor (**L**); 200× magnification. (**D**–**I**) Cell viability (CCK-8 assay) and intracellular total iron, Fe^2+^, GSH, GSSG, and GSH/GSSG levels in erastin-treated trophoblast cells transfected with the miR-205_R-1 mimic. (**J**) Transfection efficiency of the miR-205_R-1 inhibitor at different concentrations (50, 100, and 200 μM) in trophoblast cells. (**M**–**R**) Cell viability (CCK-8 assay), and intracellular total iron, Fe^2+^, GSH, GSSG, and GSH/GSSG levels of erastin-treated trophoblast cells transfected with the miR-205_R-1 inhibitor. Data are presented as mean (SD); * represents *p* < 0.05; ** represents *p* < 0.01; ns: *p* > 0.05.

**Figure 4 antioxidants-13-00451-f004:**
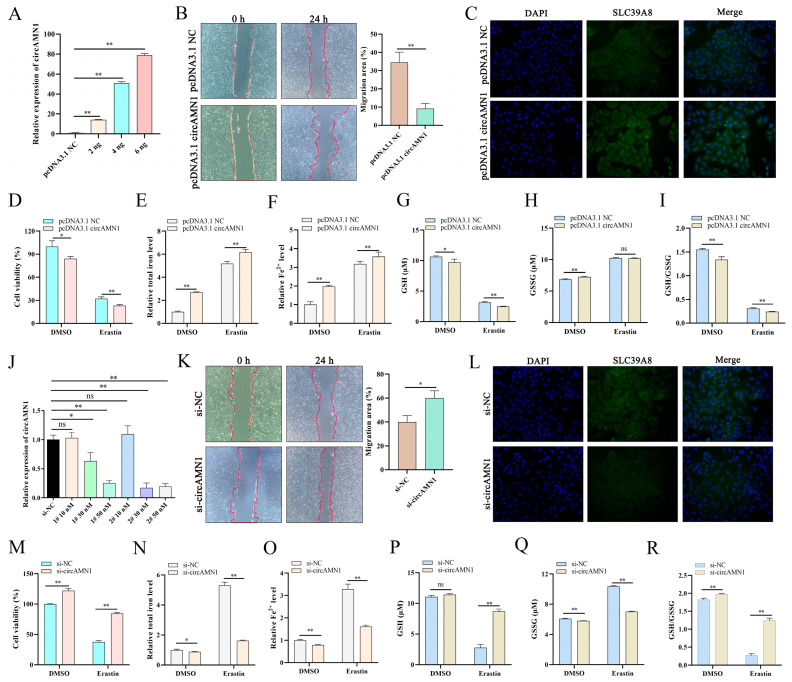
circAMN1 regulates ferroptosis in trophoblast cells. (**A**) Transfection efficiency of different concentrations of pcDNA3.1 circAMN1 (2, 4, and 6 ng) in trophoblast cells. (**B**,**K**) Wound-healing assay of trophoblast cells transfected with 4 ng of pcDNA3.1 circAMN1 (**B**) or 30 nM si-circAMN1 (**K**), imaged at 0 and 24 h following wound infliction. The migration area was calculated using ImageJ; 40× magnification. (**C**,**L**) Immunofluorescence assay of trophoblast cells transfected with pcDNA3.1 circAMN1 (**C**) or si-circAMN1 (**L**); 200× magnification. (**D**–**I**) Cell viability (CCK-8 assay) and intracellular total iron, Fe^2+^, GSH, GSSG, and GSH/GSSG levels of erastin-treated trophoblast cells transfected with pcDNA3.1 circAMN1. (**J**) Transfection efficiency of si-circAMN1 #1 and #2 at different concentrations (10, 30, and 50 nM) in trophoblast cells, indicated by circAMN1 expression. (**M**–**R**) Cell viability (CCK-8 assay) and intracellular total iron, Fe^2+^, GSH, GSSG, and GSH/GSSG levels of erastin-treated trophoblast cells transfected with si-circAMN1. Data are presented as mean (SD); * represents *p* < 0.05; ** represents *p* < 0.01; ns: *p* > 0.05.

**Figure 5 antioxidants-13-00451-f005:**
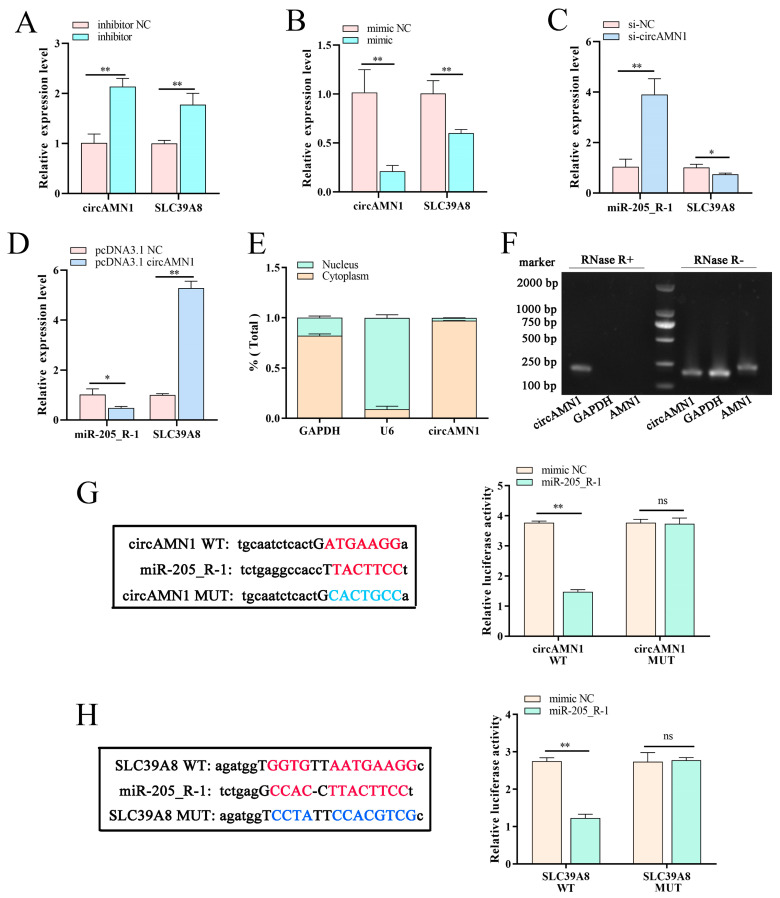
Targeting relationship among circAMN1, SLC39A8, and miR-205_R-1. *circAMN1* and *SLC39A8* mRNA expression in trophoblast cells following transfection with an miR-205_R-1 inhibitor (**A**) or mimic (**B**). *miR-205_R-1* and *SLC39A8* mRNA expression in trophoblast cells following transfection with si-circAMN1 (**C**) or pcDNA3.1 circAMN1 (**D**). (**E**) circAMN1 expression in the cytoplasm and nuclei of trophoblast cells; *GAPDH* and *U6* were used as cytoplasmic and nuclear controls, respectively. (**F**) PCR amplification of *AMN1* and *circAMN1* mRNA in RNase R-treated trophoblast cells; *GAPDH* served as a negative control. Binding site between *circAMN1* and *miR-205_R-1* (**G**) and *SLC39A8* and *miR-205_R-1* (**H**); the red base represents the wild-type binding site and the blue base represents the mutant binding site; a corresponding luciferase reporter assay was performed to detect binding between miR-205_R-1 and circAMN1 (**G**) and miR-205_R-1 and SLC39A8 (**H**). Data are presented as mean (SD); * represents *p* < 0.05; ** represents *p* < 0.01; ns: *p* > 0.05.

**Figure 6 antioxidants-13-00451-f006:**
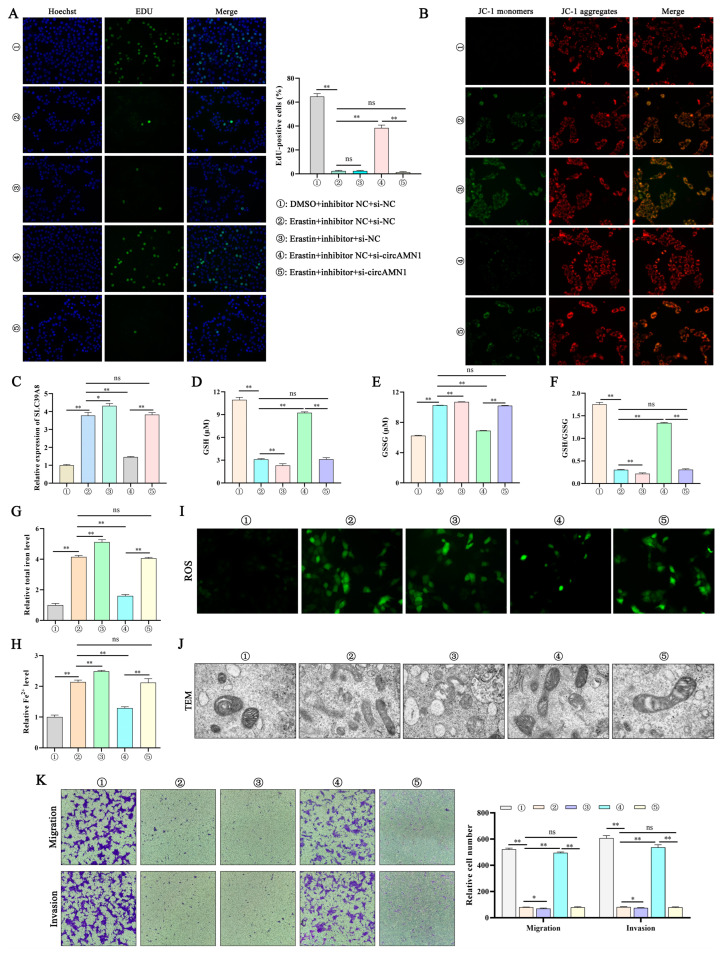
circAMN1 acts as an miR-205_R-1 sponge to regulate ferroptosis. (**A**–**K**) Erastin-treated trophoblast cells, co-transfected with si-circAMN1 and an miR-205_R-1 inhibitor. (**A**) Cell proliferation (EdU assay) and number of EdU-positive cells; 200× magnification. (**B**) Mitochondrial membrane potential; 200× magnification. (**C**) *SLC39A8* mRNA expression determined using quantitative PCR. (**D**–**H**) Intracellular GSH, GSSG, GSH/GSSG, total iron, and Fe^2+^ levels. (**I**,**J**) Reactive oxygen species levels (200× magnification) and the corresponding transmission electron microscopy (8000× magnification) of trophoblast cells. (**K**) Migration and invasion of trophoblast cells determined using the Transwell assay; 40× magnification. Data are presented as mean (SD); * represents *p* < 0.05; ** represents *p* < 0.01; ns: *p* > 0.05.

**Figure 7 antioxidants-13-00451-f007:**
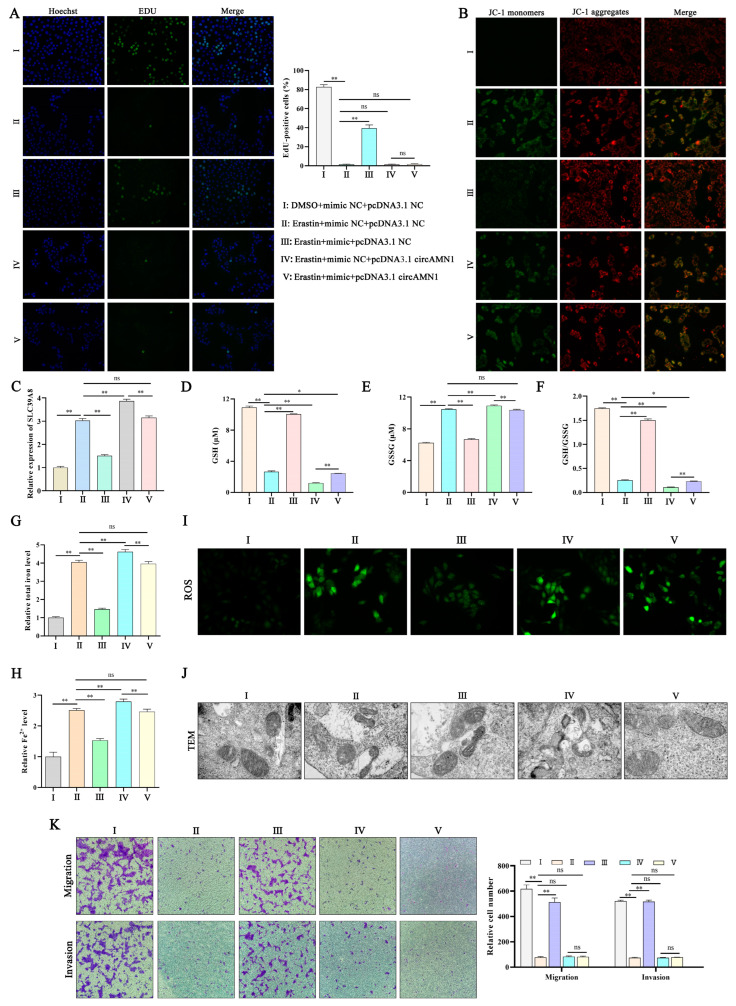
circAMN1 regulates the miR-205_R-1/SLC39A8 axis to mediate ferroptosis. (**A**–**K**) Erastin-treated trophoblast cells co-transfected with pcDNA3.1 circAMN1 and an miR-205_R-1 mimic. (**A**) Proliferation (EdU assay) and number of EdU-positive cells; 200× magnification. (**B**) Mitochondrial membrane potential; 200× magnification. (**C**) *SLC39A8* mRNA expression determined using quantitative PCR. (**D**–**H**) Intracellular GSH, GSSG, and GSH/GSSG, total iron, and Fe^2+^ levels. (**I**,**J**) reactive oxygen species levels (200× magnification) and the corresponding transmission electron microscopy of trophoblast cells (8000× magnification). (**K**) Migration and invasion of trophoblast cells determined using the Transwell assay; 40× magnification. Data are presented as mean (SD); * represents *p* < 0.05; ** represents *p* < 0.01; ns: *p* > 0.05.

## Data Availability

The datasets supporting the conclusions of this article are available in the Gene Expression Omnibus repository as follows: GSE214588 (https://www.ncbi.nlm.nih.gov/geo/query/acc.cgi?acc=GSE214588), accessed on 18 October 2022 and GSE248362 (https://www.ncbi.nlm.nih.gov/geo/query/acc.cgi?acc=GSE248362), accessed on 21 November 2023.

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
