# Peer review of "circAMN1-Mediated Ferroptosis Regulates the Expulsion of Placenta in Trophoblast Cells"

_antioxidants, 2024, doi:10.3390/antiox13040451_

Round 1

Reviewer 1 Report

In this study the authors investigated circAMN1 regulates the expulsion of the placenta through a process that involved ferroptosis. The experimental designed is well laid out. The methodology in some cases as presented does not allow for reproducibility need further description see below for specificity. The results and conclusions are adequate, and the manuscript has potential for high impact. Below are major concerns that the authors may consider.

1.       The title can be modified to “circAMN1 mediated ferroptosis regulates the expulsion of placenta in trophoblast cells”

2.       The following methods (2.5, 2.6, 2.7, 2.8, 2.11, 2.12 and 2.14) need additional short description.

3.       There is no uniformity on the significant signs on the bars in some figure’s lines are drawn across two bars while others have signs placed on the bars. Also, some bars have no signs on them.

In this study the authors investigated circAMN1 regulates the expulsion of the placenta through a process that involved ferroptosis. The experimental designed is well laid out. The methodology in some cases as presented does not allow for reproducibility need further description see below for specificity. The results and conclusions are adequate, and the manuscript has potential for high impact. Below are major concerns that the authors may consider.

1.       The title can be modified to “circAMN1 mediated ferroptosis regulates the expulsion of placenta in trophoblast cells”

2.       The following methods (2.5, 2.6, 2.7, 2.8, 2.11, 2.12 and 2.14) need additional short description.

3.       There is no uniformity on the significant signs on the bars in some figure’s lines are drawn across two bars while others have signs placed on the bars. Also, some bars have no signs on them.

Author Response

Dear Reviewer:

Thank you for your comment on our manuscript titled “The circAMN1/miR-205_R-1/SLC39A8 axis regulates the expulsion of the placenta by mediating ferroptosis in trophoblast cells” (ID: antioxidants-2929995). Those comments were valuable and helpful for revising and improving our paper, and provided important guidance for our research. We have considered comments carefully and have revised the manuscript accordingly. The revised portions are marked in yellow in the manuscript. The main corrections in the paper and responses to the reviewer’s comments are shown below.

Responds to Reviewer’s comments:

1.Reviewer’s comment: The title can be modified to “circAMN1 mediated ferroptosis regulates the expulsion of placenta in trophoblast cells”

Response: We have made correction according to the Reviewer’s comments and highlighted it in yellow in the manuscript.

2.Reviewer’s comment: The following methods (2.5, 2.6, 2.7, 2.8, 2.11, 2.12 and 2.14) need additional short description.

Response: We have re-written this part according to the Reviewer’s suggestion and highlighted it in yellow in the manuscript.

3.Reviewer’s comment: There is no uniformity on the significant signs on the bars in some figure’s lines are drawn across two bars while others have signs placed on the bars. Also, some bars have no signs on them.

Response: As Reviewer suggested that we have unified the significant signs.

We have carefully revised the manuscript. These changes did not influence the content or framework of the paper. We appreciate for Reviewers’ warm work earnestly, and hope that the correction will meet with approval.

Once again, thank you for your comments and suggestions.

                                                                Yours sincerely,

                                                                       Chen Lv

Reviewer 2 Report

Dear authors,

Your article is quite interesting with many information related to this topic. I have only one suggestion. I believe that you presented many information in each figure and it is a bit chaotic. In my opinion, if you could improve this point, it would help the readers to better understand your research work.

Kind regards,

Redesign images

Author Response

Dear Reviewer:

Thank you for your comment on our manuscript titled “The circAMN1/miR-205_R-1/SLC39A8 axis regulates the expulsion of the placenta by mediating ferroptosis in trophoblast cells” (ID: antioxidants-2929995). Those comments were valuable and helpful for revising and improving our paper, and provided important guidance for our research. We have considered comments carefully and have revised the manuscript accordingly. The main corrections in the paper and responses to the reviewer’s comments are shown below.

Responds to Reviewer’s comments:

Reviewer’s comment: Redesign images

Response: As Reviewer suggested that we have remade the image.

We have carefully revised the manuscript. These changes did not influence the content or framework of the paper. We appreciate for Reviewers’ warm work earnestly, and hope that the correction will meet with approval.

Once again, thank you for your comments and suggestions.

                                                                    Yours sincerely,

                                                                            Chen Lv

Reviewer 3 Report

I found the present work quite exaustive, including both useful in vivo and in vitro approaches. Well done.

I can however suggest to rephrase the conclusive sentences regarding the potential application of circAMN1 and miR-205_R-1 screening: the hypotesis of the authors will certainly need to be validated by extensive in field study, considering different breed, breeding conditions and other relevant sources of confounding variance.

Regarding all bioinformatics analysis I suggest the author to add as supplementary data a file with a resume of all RNAseq pipelines and related versions of R packages used in the study, reference genome assembly, aligner, etc.. These information are nowadays crucial to gather reproducible bioinformatics, in order to keep track of how reported outputs were produced, avoiding/reducing manual data manipulation steps and to archive the exact versions of all external programs used and version control of all custom scripts (if any).

None

Author Response

Dear Reviewer:

Thank you for your comment on our manuscript titled “The circAMN1/miR-205_R-1/SLC39A8 axis regulates the expulsion of the placenta by mediating ferroptosis in trophoblast cells” (ID: antioxidants-2929995). Those comments were valuable and helpful for revising and improving our paper, and provided important guidance for our research. We have considered comments carefully and have revised the manuscript accordingly. The revised portions are marked in yellow in the manuscript. The main corrections in the paper and responses to the reviewer’s comments are shown below.

Responds to Reviewer’s comments:

Reviewer’s comment: I found the present work quite exaustive, including both useful in vivo and in vitro approaches. Well done.

I can however suggest to rephrase the conclusive sentences regarding the potential application of circAMN1 and miR-205_R-1 screening: the hypotesis of the authors will certainly need to be validated by extensive in field study, considering different breed, breeding conditions and other relevant sources of confounding variance.

Regarding all bioinformatics analysis I suggest the author to add as supplementary data a file with a resume of all RNAseq pipelines and related versions of R packages used in the study, reference genome assembly, aligner, etc.. These information are nowadays crucial to gather reproducible bioinformatics, in order to keep track of how reported outputs were produced, avoiding/reducing manual data manipulation steps and to archive the exact versions of all external programs used and version control of all custom scripts (if any).

Response: We have made correction according to the Reviewer’s comments and highlighted it in yellow in the manuscript. All bioinformatics analysis data has been organized and supplementary data will be uploaded again.

We have carefully revised the manuscript. These changes did not influence the content or framework of the paper. We appreciate for Reviewers’ warm work earnestly, and hope that the correction will meet with approval.

Once again, thank you for your comments and suggestions.

                                                                Yours sincerely,

                                                                        Chen Lv
